# Randomised double-blind phase 3 clinical study testing impact of atorvastatin on prostate cancer progression after initiation of androgen deprivation therapy: study protocol

Aino Siltari [1,2] Jarno Riikonen,[3] Juha Koskimäki,[3] Tomi Pakarainen,[3]
Otto Ettala [4] Peter Boström,[4] Heikki Seikkula,[5] Andres Kotsar,[6] Teuvo Tammela,[1]
Mika Helminen,[7] Paavo V Raittinen,[8] Terho Lehtimäki,[9] Mikkel Fode,[10]
Peter Østergren,[10] Michael Borre,[11] Antti Rannikko,[12] Timo Marttila,[13]
Arto Salonen,[14] Hanna Ronkainen,[15] Sven Löffeler [16] Teemu J Murtola [3]

For numbered affiliations see end of article.

**Correspondence to**
Dr Teemu J Murtola;
teemu.murtola@tuni.fi

## ABSTRACT

**Introduction** Blood cholesterol is likely a risk factor for prostate cancer prognosis and use of statins is associated with lowered risk of prostate cancer recurrence and progression. Furthermore, use of statins has been associated with prolonged time before development of castration resistance (CR) during androgen deprivation therapy (ADT) for prostate cancer. However, the efficacy of statins on delaying castration-resistance has not been tested in a randomised placebo-controlled setting. This study aims to test statins' efficacy compared to placebo in delaying development of CR during ADT treatment for primary metastatic or recurrent prostate cancer. Secondary aim is to explore effect of statin intervention on prostate cancer mortality and lipid metabolism during ADT.

**Methods and analysis** In this randomised placebo-controlled trial, a total of 400 men with de novo metastatic prostate cancer or recurrent disease after primary treatment and starting ADT will be recruited and randomised 1:1 to use daily 80 mg of atorvastatin or placebo. All researchers, study nurses and patients will be blinded throughout the trial. Patients are followed until disease recurrence or death. Primary outcome is time to formation of CR after initiation of ADT. Serum lipid levels (total cholesterol, high-density lipoprotein (HDL), low-density lipoprotein (LDL) and trigyserides) are analysed to test whether changes in serum cholesterol parameters during ADT predict length of treatment response. Furthermore, the trial will compare quality of life, cardiovascular morbidity, changes in blood glucose and circulating cell-free DNA, and urine lipidome during trial.

**Ethics and dissemination** This study is approved by the Regional ethics committees of the Pirkanmaa Hospital District, Science centre, Tampere, Finland (R18065M) and Tarto University Hospital, Tarto, Estonia (319/T-6). All participants read and sign informed consent form before study entry. After publication of results for the primary endpoints, anonymised summary metadata and statistical code will be made openly available. The data will not

## Strengths and limitations of this study

► First randomised placebo-controlled phase 3 clinical study on effects of atorvastatin on prostate cancer progression during androgen deprivation therapy treatment.
► Multicentre study in Finland, Denmark, Estonia and Norway.
► As a limiting factor, only minority of prostate cancer cases are advanced and large proportion of potential participants are ineligible due to using statins already. Thus, study enrolment will take several years.

include any information that could make it possible to identify a given participant.

**Trial registration number** Clinicaltrial.gov: NCT04026230, Eudra-CT: 2016-004774-17, protocol code: ESTO2, protocol date 10 September 2020 and version 6.

## INTRODUCTION

Prostate cancer (PCa) is the most common cancer in Finnish men and a major public health burden causing annually around 900 cancer deaths in Finland and the yearly costs caused by the disease are estimated to reach 180 million euros.[1 2] However, not all prostatic malignancies are lethal; only 10%–20% of tumours advance to metastases and eventually into a fatal stage. Advanced PCa is treated by androgen deprivation therapy (ADT). Eventually, however, PCa progress despite the ADT treatment and forms state called castration resistance. Thus, castration resistance is clinically defined as a moment when PCa no longer responds to ADT treatment. Median time to castration resistance is 12–15 months

and 45 months to death in patients who have started ADT for metastatic PCa.[3–7] In men who start ADT for disease recurrence after primary treatment, median failure-free survival time is 33 months and 70 months to death.[3]

Blood cholesterol is likely a risk factor for PCa prognosis; risk of disease recurrence after primary treatment is significantly elevated in men with hypercholesterolaemia.[8] Laboratory studies have demonstrated the importance of cholesterol for PCa cell growth.[9 10] Furthermore, upregulation of intracellular lipid production and ensuing lipid accumulation is essential in surviving hypoxic tumour microenvironment.[11] In PCa cells, it also assists in evading host-tumour immune response.[12] Also, upregulation of intracellular cholesterol production appears to be central for development of castration resistance.[13]

Lipid metabolism is emerging as a new risk factor for PCa progression. The role of cholesterol is important in development of castration resistance, and inhibition of intracellular lipid production interferes with androgen receptor (AR) signalling essential for PCa progression.[13–15] Cholesterol is precursor for intracellular androgen production, a central mechanism for PCa cells to overcome ADT.[13 15] A recent study suggests that some of the serum basic lipid parameters may have increased association with development of castration resistance and metastasis especially in statin-naïve patients.[16] Clarification of the role of lipid metabolism during ADT will likely provide new tools for control of disease progression and PCa treatment.

Use of cholesterol-lowering statin drugs is associated with lowered risk of PCa recurrence and progression; risk of PCa death is reduced by 30% compared with the non-users.[14 17 18] The anticancer effect may be specifically against progression of the disease and the mechanisms driving it.[19] Statins inhibit the cholesterol-synthesising mevalonate pathway, which is active in PCa cells.[10 20] Besides cholesterol, this pathway also produces isoprenoid proteins which are critical for regulation of cell growth and other central cellular control processes.[20] Furthermore, steroid hormones such as testosterone are metabolised from cholesterol, thus statins appear to target androgen metabolism, another crucial pathway for PCa growth.[21 22]

Statin use has been reported to prolong efficacy of ADT in PCa as it has been reported to prolong the efficacy of ADT for 8–10 months.[23 24] Furthermore, statins have been linked to a prolonged response to androgen-signalling targeted drugs abiraterone and enzalutamide used in management of castration-resistant disease.[25 26]

Statins have not been found to affect PCa mortality in trials testing their efficacy in secondary prevention of cardiovascular disease.[27] However, in these trials, cancer was often an exclusion criterion. On the other hand, it may be that statin treatment has more impact on hormone-dependent cancers which are underrepresented in these studies. In a randomised clinical trial focusing on PCa patients 80 mg of atorvastatin has been found to reduce tumour proliferation activity compared with placebo after minimum exposure of 21 days.[28] Similar results were seen after treatment with fluvastatin, although in a non-randomised and uncontrolled setting.[29] Nevertheless, clinical efficacy of statins in preventing progression of PCa has not been tested in a randomised placebo-controlled setting. Also, a recent post-hoc study of randomised clinical trial concluded that statin use was associated with decreased overall and PCa-specific mortality in men with ADT.[30] Therefore, it is important to do a trial testing effects of statins specifically in PCa patients.

## Study objective

Primary objective for this phase III randomised double-blind placebo-controlled trial is to explore whether intervention with atorvastatin delays PCa progression that is, development of castration resistance compared to placebo during ADT for metastatic or recurrent PCa. Secondary objectives include exploring whether atorvastatin lowers PCa-specific or overall mortality compared with placebo, and to demonstrate whether changes in serum lipid parameters predict disease recurrence and occurrence of adverse tumour genomic traits predicting castration resistance among PCa patients during ADT.

## METHODS AND ANALYSIS

### Study setting

Study flow, study settings and other information are presented in figure 1 and table 1. The study recruitment target is 400 participants who start ADT as primary management of de novo metastatic PCa or as secondary management for PCa recurrence after localise treatment. Participants can be high-risk M0 or M1 stage, main inclusion criterion is that long-term ADT treatment is started. These men will be randomised 1:1 (200+200) to receive either 80 mg of atorvastatin daily or placebo until disease recurrence that is, development of castration resistance, death or maximum of ten years. Sample size is based on a power calculation from a previous retrospective study.[23]

The study will be carried out in collaboration between urological departments of University Hospitals and central hospitals in Finland, the Herlev University Hospital in Denmark, the Tartu University Hospital in Estonia and Vestfold Hospital Trust in Norway (table 1).

Study data are collected and managed using REDCap electronic data capture tools hosted at Tampere University.[31 32] Only the primary investigators, study nurses and registered study coinvestigator from each participating site will have access to the platform. All laboratory results, symptoms and results from imaging studies done at the discretion of the attending clinicians are recorded in the database. All clinical decisions besides the study drug, for example, use of early chemotherapy, abiraterone or other drugs in adjunct with ADT will be up to the discretion of the attending clinician and allowed but will also be recorded in the REDCap database.

Follow-up is continued until development of castration resistance, death or maximum of 10 years. Participants

**Figure 1** Study flow with inclusion and exclusion criteria. ADT, androgen deprivation therapy.

are given the opportunity to carry on with the intervention even after development of castration resistance to observe effects on survival. Unblinding will be performed after recruitment target has been reached and all participants have been followed for minimum of 12 months.

Castration resistance is defined as either PSA progression (three consecutive rises of PSA measured at least 1 week apart with two >50% increases over the nadir and PSA >2 ng/mL) or radiological progression (appearance of two or more lesions in bone scan or soft tissue enlargement as per RECIST criteria) while serum testosterone is at the castrate level (<50 ng/mL or 1.7 nmol/L).

For men who initiate statin use during the study period for clinical indications, the study drug is dropped but the study follow-up is continued. These men will be included in the final analysis according to the intention-to-treat principle within their allocated study arm.

In case of intolerable side effects as judged by either the participant or the attending physician, the study drug is stopped, and these men will be analysed according to the intention-to-treat principle.

Participants who discontinue follow-up or deviate from the study protocol for any reason will be given the chance to remain or return to follow-up to allow intention-to-treat analyses.

## Inclusion and exclusion criteria

Inclusion criteria for participants are histopathologically confirmed metastatic adenocarcinoma of the prostate or high-risk M0 stage recurrent PCa for which androgen deprivation or antiandrogen therapy is initiated no longer than 3 months before recruitment, willingness to participate and signing of informed consent (figure 1).

Exclusion criteria for participants are regular statin use at the time of recruitment or within 6 months of it, previous adverse effects during statin therapy, familial hypercholesterolaemia or very high total cholesterol (9.3 mmol/L or above), clinically significant renal (serum creatinine above 170 µmol/L) or liver insufficiency (serum alanine aminotransferase more than two times above the upper limit of normal range), and use of drugs that may interact with statins (St John's Wort, HIV protease inhibitors, ciclosporin, macrolide antibiotics, fucidic acid, phenytoin, carbamazepine, dronedarone or oral antifungal medication) (figure 1).

## Study endpoints

Primary endpoint is the time to disease progression after starting ADT/antiandrogen therapy. Secondary endpoints are (1) PCa-specific mortality and overall survival, (2) change in serum cholesterol during the intervention and its role in predicting time to disease recurrence in the placebo arm, (3) occurrence of adverse tumour traits predicting development of castration resistance in circulating cell free DNA, (4) changes in fasting blood glucose during ADT, (5) occurrence of cardiovascular events during ADT and (6) Quality of life (QoL) during ADT.

**Table 1** Study setting of Impact of atorvastatin on prostate cancer after initiation of androgen deprivation therapy clinical trial

| Study settings | |
| --- | --- |
| Primary registry and trial identifying no | Clinicaltrials.gov NCT04026230 |
| Date of registration in primary registry | 19 July 2019 |
| Secondary identifying numbers | Eudra-CT: 2016-004774-17, Protocol code: ESTO2 |
| Source(s) of monetary or material support | Tampere University Hospital, Finland |
| Primary sponsor | Tampere University Hospital, Finland |
| Secondary sponsor(s) | Helsinki University Hospital,Turku University Hospital, Central Finland Central Hospital, Kuopio University Hospital, Oulu University Hospital, Finland, Herlev Hospital, Denmark, University Hospital Tarto, Estonia, Vestfold Hospital Trust, Tønsberg, Norway |
| Contact for public queries | Tampere University Hospital, Teemu Murtola, MD, PhD |
| Contact for scientific queries | Tampere University Hospital, Teemu Murtola, MD, PhD |
| Public title | Impact of atorvastatin on prostate cancer progression after initiation of androgen deprivation therapy |
| Scientific title | Impact of atorvastatin on prostate cancer progression after initiation of androgen deprivation therapy— lipid metabolism as a novel biomarker to predict prostate cancer progression—phase 3, double-blind randomised clinical trial FinnProstata XV |
| Countries of recruitment | Finland, Denmark, Estonia, Norway |
| Health condition studied | Metastatic or recurrent prostate cancer |
| Intervention | Active comparator: Capsules of atorvastatin 80 mg, Placebo comparator: Similar capsules as in the atorvastatin arm, but without the active ingredient |
| Key inclusion and exclusion criteria | Inclusion criteria: Histopathologically confirmed metastatic adenocarcinoma of the prostate for which androgen deprivation or antiandrogen therapy is initiated no longer than 3 months before as the primary treatment Willingness to participate and signing of informed consent exclusion criteria: Statin use at the time of recruitment or within 6 months of it, Previous adverse effects during statin therapy, familial hypercholesterolaemia or very high total cholesterol, clinically significant renal or liver insufficiency, use of drugs that may interact with statins |
| Sexes eligible for study: | Male |
| Accepts healthy volunteers | No |
| Study type | Interventional, allocation: randomised, Intervention model: parallel assignment with 1:1 allocation ratio, Masking: double blind (subject, caregiver, investigator, outcomes assessor), Primary purpose: prevention, phase III |
| Date of first enrolment | July 2019 |
| Target sample size | 400 |
| Recruitment status | Recruiting |
| Primary outcome(s) | Castration resistance |

Continued

**Table 1** Continued

| Study settings | |
| --- | --- |
| Key secondary outcomes | Lipid levels, Prostate cancer mortality and overall survival, Circulating cell free DNA, Fasting blood glucose, Occurrence of cardiovascular events during ADT, Quality of life |

ADT, androgen deprivation therapy.

## Study flow

Study flow and schedule is presented in figure 1 and table 2. Patients are recruited at urology outpatient clinic at urologists' visits. If the inclusion criteria are met and the exclusion criteria are not and the participant signs informed consent form, he is given a study number (ranging between E-001 to E-400) and he receives either 80 mg atorvastatin or placebo according to the study arm randomly allocated for the study number. No blocking or other restrictions will be implemented for randomisation. Only the national study coordinator in each country will be able to see which study arm the participant has been randomised to. The participant will then receive first dose of the respective study drug randomised to his study number. The drug boxes and capsules containing atorvastatin and placebo will be identical in appearance. All researchers, nurses and study participants will remain blinded to the allocation sequence until termination of the study and closure of the study data.

Control visits are scheduled at 6-month intervals to suit common clinical practice. Control visits are done by a clinician, complemented with measurements done by a research nurse. For each control visit the participant returns remaining capsules from the previously delivered ration and will receive a new one according to his study number. Number of the remaining capsules will be counted and saved in the REDCap, trial database, to monitor compliance. In every visit, serum PSA (ng/mL), lipid levels (total cholesterol, high-density lipoprotein (HDL), low-density lipoprotein (LDL) and triglycerides, mmol/L), alkaline phosphatase (U/L), creatinine (µmol/L), and fasting glucose (mmol/L) will be measured (table 2) and the participant is asked about symptoms that are suggestive of metastases and about possible adverse effects of the study drug. Serum creatine kinase (U/L) and alanine aminotransferase (U/L) is also measured at each visit.

Blood samples are taken and stored at 6-month intervals to monitor changes in tumour markers and occurrence of adverse tumour characteristics after initiation of ADT. QoL during the intervention will be charted once a year using validated WHOQOL-BREF QoL questionnaire.[33]

Follow-up is continued until development of castration resistance, drop out from the study, death or maximum of 10 years. Participants developing castration resistance are given an opportunity to carry on with their assigned

**Table 2** Study flow and timing of the laboratory and blood sample collection

| Timepoint | Enrolment | Allocation | Follow-up‡ | | | |
|---|---|---|---|---|---|---|
| | | 0 | 6 months | 12 months | 18 months | 24 months |
| Enrolment: | | | | | | |
| Eligibility screen | x | | | | | |
| Informed consent | x | | | | | |
| Allocation | | x | | | | |
| Interventions:* | | | | | | |
| Atorvastatin | | | x | x | x | x |
| Placebo | | | x | x | x | x |
| Assesments: | | | | | | |
| Serum PSA (ng/mL) | x | | x | x | x | x |
| Alkaline phosphatase (U/L) | x | | x | x | x | x |
| Creatinine (µmol/L) | x | | x | x | x | x |
| Serum creatine kinase (U/L) | x | | x | x | x | x |
| Alanine aminotranspherase (U/L) | x | | x | x | x | x |
| Fasting blood glucose (mmol/L) | x | | x | x | x | x |
| Lipid levels (total cholesterol, LDL, HDL and triglycerides) (mmol/L) | x | | x | x | x | x |
| Cell free DNA (subgroup)† | x | | | | | |
| QoL | x | | | x | | x |

*Allocation 1:1 to have eighter 80 mg atorvastatin or placebo daily
†Cell free DNA is measured before allocation and after occurrence of castration resistance.
‡Follow-up until disease recurrance, death or max. 10 years
HDL, high-density lipoprotein; LDL, low-density lipoprotein; PSA, prostate-specific antigen; QoL, quality of life.

treatment as statins have been linked to longer cancer-specific survival. Unblinding will be performed after recruitment target has been reached and all participants have been followed for minimum of 12 months.

PCa treatment apart from the study drug will be up to the discretion of the attending clinician. All laboratory results, symptoms and possible imaging results will be recorded in the REDCap database. All clinical decisions regarding early chemotherapy and imaging will be made on the discretion of the clinician and recorded in the REDCap database.

### Stored blood samples
Separate whole blood (for RNA and DNA isolation), plasma and serum samples are taken at each control visit at 6-month intervals and stored for mass-spectrometric and nuclear MR-based determination of serum lipidome. These samples will also be used for RNA and whole genome sequencing of mutations and genetic modifications predicting disease recurrence and metastasis, such as BRCA1/2, ERG, MYC, TP53, ATM, PTEN and AR splice variants.[34 35] Also hypoxia markers will be measured.[36]

For participants with confirmed metastases in the bone or soft tissues, amount of cell free DNA from the plasma will be measured before ADT initiation and again at castration resistance development for sequencing and detection of genetic modifications predicting metastases.[34 35] Selected patients are also imaged with positron emission tomography (PET) scan using 18F-2-nitroimidazolpentafluoropropylace tamide (EF5)[37] to monitor hypoxia and fluorine-18-fluorodeoxyglucose positron emission tomography (FDG-PET) to evaluate immune responses[38] in the primary tumour and the metastases during the atorvastatin intervention.

### Sample size calculation
In a cohort study by Harshman et al,[23] among men starting ADT, 58% of the statin users and 75% of non-users of statins progress to castration resistance during median 5.8 years. We used these crude percentages for sample size calculations.

With alpha and beta values of 0.05 and 0.20 (power=0.80), sample size of 400 men will be enough to detect a risk difference with HR 0.65 (figure 2). We assume 10% drop-out rate in each study arm. The programme 'PS—Power and sample size, V.3.1.2' was used for the calculation.

The median time to disease progression is assumed to be 12–15 months for patients with de novo metastatic disease and 33 months for patients recurring

**Sample size as a function of HR, power = 0.8**

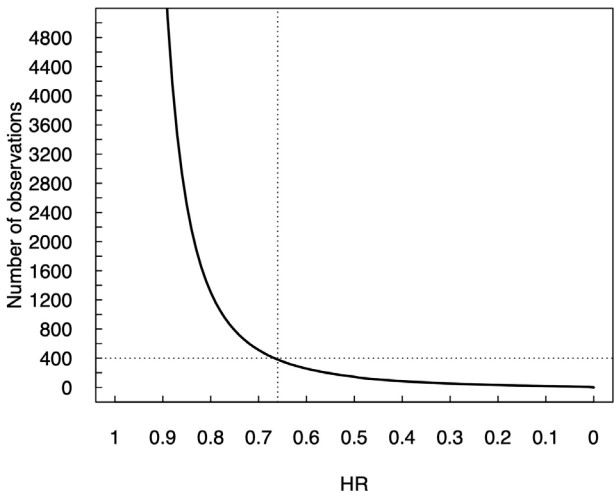

**Figure 2** Sample size estimation with power 0.8 as a function of HR. Calculation was made based on study by Harshman *et al.*[23]

after primary therapy.[3–7] Therefore, the intervention will continue until castration resistance, death, or for maximum of 10 years. Post hoc follow-up will continue after the intervention. A 3 months' difference in time to castration resistance between the study arms will be considered clinically significant.

It has to be noted that the number of studies published on this topic is low and, in the absence of randomised evidence on this topic, the power calculation is based on results of epidemiological study prone to residual bias. Therefore, an interim analysis will be performed after the first 100 participants have met the primary endpoint, that is, progressed into castration resistance. Without lifting the blinding, the power calculations are repeated to verify whether the chosen sample size has adequate statistical power to detect statistically significant difference of the effect size observed between the randomisation groups at that time. If the calculations suggest that a larger sample size is needed for the trial, the study protocol will be amended with updated recruitment target.

### Statistical methods

Distribution balance of PCa clinical characteristics such as TNM stage at diagnosis, Gleason score and PSA at time of diagnosis will be presented in a patient characteristics table, stratified by study arms. Categorical variables are displayed as absolute count frequencies and relative frequencies, that is, proportion within study arm. Continuous variables are presented as median and IQR. Minimum and maximum values are not shown to ensure anonymity of the participants.

Recurrence-free survival, that is, time from ADT initiation to castration resistance, will be analysed by using Kaplan-Meier (KM) estimator, stratified by study arm. The statistical significance of the difference between the survival estimates is tested with log-rank test. No adjustments will be used for KM estimator as the study intervention is randomised. Risk of death, as well as risk for occurrence of adverse tumour traits among men receiving atorvastatin compared with men receiving placebo will be estimated using Cox proportional hazards regression.

Univariate analysis of changes in serum lipid levels is performed by calculating the difference after—before for each participant. The statistical significance of the difference between the study arms is analysed by Mann-Whitney U test (or by t-test if under normality). These results are complemented with boxplots. Multivariable analysis for difference in serum lipid levels between the study arms are performed by fitting a logistic regression model such that the study arm is used as the response variable whereas lipid level differences are used as predictors (ie, each lipid level difference as an independent predictor). Lipid levels are expected to display heavy multicollinearity; therefore, variance inflation factors (VIF) are calculated for each coefficient afterwards, and possible adjustments made by dropping predictors with extreme VIFs (>10).

The occurrence of adverse tumour genomic traits will be analysed by calculating the proportion of participants developing adverse tumour genomic traits. The statistical significance between the proportions is tested by proportions test.

The WHOQOL-BREF patient-reported outcome (PRO) of the quality-of-life is analysed by tabulating absolute and relative frequencies. $\chi^2$ test for homogeneity is used to test the statistical significance of the difference between the study arms of each PRO item.

Subgroup analyses will be performed stratified by type of ADT or antiandrogen therapy initially selected; GnRH agonists/antagonists, antiandrogen, orchiectomy or other. Men who start with antiandrogen therapy often change to GnRH agonists/antagonists or orchiectomy when PSA increases during ADT. This is allowed and PSA recurrence during antiandrogen therapy is not considered as disease progression as serum testosterone needs to be at castrate level as well for meeting the primary endpoint.

Another subgroup analysis will be stratified by whether or not the participants have received local therapy (radiation or surgery) for PCa in addition to ADT and by type of additional therapy participant has received; docetaxel, abiraterone, enzalutamide or apalutamide at hormone-sensitive stage in addition to ADT.

All analyses will include only participants with available data, imputations will not be used. Detailed statistical analysis plan is published before final data analysis. All analyses are performed using SPSS V.27, Stata V.17 and R V.4 statistical softwares.

## Quality control

The study steering committee includes investigators from each participating recruitment centre. The steering committee will meet twice a year to oversee trial progression of the recruitment, integrity of collected data and discuss possible protocol amendments.

External trained study monitors independent from the study sponsors ensure the data quality and good clinical practice by making regular yearly check-up visits to participating study centres. This will be separately arranged in each participating country.

## Reporting and registering of adverse effects

The participants are advised to contact the researchers in case of suspected adverse effect related to the study drug. If the side-effects are intolerable the study drug will be discontinued. In uncertain cases serum levels of creatine kinase (U/L), alanine aminotranspherase (U/L) and creatinine (µmol/L) are checked to see if any of these have changed considerably compared with the baseline. If the laboratory tests are not normal, or if the participant or the attending physician finds the side effects alarming the study drug will be discontinued. Study database records specifically most common adverse effects muscular pain, elevated fasting blood glucose (>6.1 mmol/L) and elevated serum creatine kinase (>280 U/L), even when they do not lead to discontinuation of the study drug.

Trial database, REDCap, includes separate question for serious adverse effects as evaluated by CTCAE criteria, V.5.0. If serious side effects are reported, REDCap asks to fulfil separate query where detailed description of the adverse effect is given according to CTCAE criteria. Study sponsor and coordinator will be automatically notified via email if serious adverse effect is reported in the database. All serious adverse effect that threatens life or health are reported to national authority (FIMEA in Finland) within 7 days of detection. If a life-threatening adverse effect is suspected blinding will be lifted for the participant in case to see whether he has received atorvastatin or placebo. Unexpected adverse effects will be reported to national authority within 15 days of detection.

## Patients and public involvement

In the design phase of the trial, primary investigator presented the study protocol in numerous public events, including patient advocacy group meetings, giving opportunity for the audience to give feedback and suggestions how to improve the study. Patient advocacy organisations of PCa patients are actively involved in finding subjects for the study through announcements in their journals and by allowing investigators frequently promote the study in their meetings. Study results will be also reported in patient organisation journals and social media sites for maximum visibility and distribution of trial outcomes.

## ETHICS AND DISSEMINATION

This study is approved by the regional ethics committee of Pirkanmaa Hospital District, Science centre, Tampere, Finland (R18065M). All participants read and sign informed consent form before study entry. The results of this study will be published in international peer reviewed journals.

The trial is ethical for the following reasons: (1) The study aims to improve treatment of metastatic PCa, which is the second most common cause of cancer death in Western countries, (2) the potential scientific and societal benefits from the project are substantial. If the study hypothesis proves to be right, it will provide an entirely new way to prevent and/or delay progression of metastatic PCa with atorvastatin, a drug that is cheap, well-tolerated, and with established cardiovascular benefits, (3) the study drug (atorvastatin) is known to be well tolerated and it is currently widely used in management of hypercholesterolaemia and cardiovascular disease. The adverse effects caused by the drug are usually mild and transient after stopping the study drug and (4); randomised, placebo-controlled design ensures the study will produce highest quality evidence that will change PCa treatment guidelines if the hypothesis proves to be right.

After publication of results for the primary endpoints, anonymised summary metadata and statistical code will be made openly available. The data will not include any information that could make it possible to identify a given participant.

### Author affiliations
[1]Faculty of Medicine and Health Technology, Tampere University, Tampere, Finland
[2]Faculty of Medicine, Pharmacology, University of Helsinki, Helsinki, Finland
[3]Department of Urology, TAYS Cancer Center, Tampere, Finland
[4]Department of Urology, University of Turku, Turku, Finland
[5]Department of Surgery, Central Finland Central Hospital, Jyvaskyla, Finland
[6]Department of Urology, Tartu University Hospital, Tartu, Tartumaa, Estonia
[7]Health Sciences, Tampere University, Tampere, Finland
[8]Department of Mathematics and Systems Analysis, Aalto University School of Science and Technology, Espoo, Finland
[9]Department of Clinical Chemistry, Tampere University, Tampere, Finland
[10]Department of Urology, Herlev and Gentofte University Hospital, Herlev, Denmark
[11]Department of Urology, Aarhus Universitetshospital, Aarhus, Denmark
[12]Department of Urology, Helsinki University and Helsinki University Hospital, Helsinki, Finland
[13]Department of Urology, Seinäjoki Central Hospital, Seinäjoki, Finland
[14]Department of Urology, Kuopio University Hospital, Kuopio, Finland
[15]Department of Urology, Oulu University Hospital, Oulu, Finland
[16]Section of Urology, Vestfold Hospital Trust, Tonsberg, Norway

**Contributors** TJM is the primarily investigator of this study and designed and planned the study with the help from all other authors. JR, TP, OE, PB, HS, AK, TT, MF, PØ, AR, TM, ASa, HR, SL and TJM will be conducting the study at the study centres. MH and PVR planned and will conduct the statistical analysis of the study. TL helped to plan and collaborate in analysis of lipidomic profile. ASi is coordinator of the trial. ASi and TJM wrote the first draft of the manuscript. All other authors further contributed to manuscript preparation and accepted the final version of the manuscript.

**Funding** This study is supported by grants from Finnish Cancer Foundation (grant numbers 024194 and 3122800563 (TJM) and 16 112 016 (TL)), Nordic Cancer Union (grant number MS793 (TJM)), Päivikki and Sakari Sohlberg Foundation (grant number 31228005901 (ASi)), competitive research grant of Pirkanmaa hospital district (grant number 9×032 (TJM) and grant X51001 (TL)),

and the Academy of Finland (grants numbers 322 098 (TL) and 3121330724 (TJM)).

**Disclaimer** None of the funders has a role in the design, management, analysis or interpretation of data, or writing and publication of results.

**Competing interests** PØ: honorarium as speaker from Ipsen A/S, Ferring Pharmaceuticals, and Astellas Pharma. MF: consultant fees and honorarium as speaker from Astellas and Ferring. HR: consultant fees from Bayer AB and honorarium as speaker from Sanofi. TJM: Consultant fees from Astellas, Janssen, speaker's honorarium from Astellas, Janssen and Sanofi, participation in scientific meetings at the expense of Ferring, Pfizer, and Sanofi, stockholder for Arocell AB. PB: consultant fees from Astellas Pharma. All other authors: No competing interests to declare.

**Patient and public involvement** Patients and/or the public were involved in the design, or conduct, or reporting, or dissemination plans of this research. Refer to the Methods section for further details.

**Patient consent for publication** Consent obtained directly from patient(s).

**Provenance and peer review** Not commissioned; externally peer reviewed.

**ORCID iDs**
Aino Siltari http://orcid.org/0000-0002-7814-427X
Otto Ettala http://orcid.org/0000-0003-1011-4602
Sven Löffeler http://orcid.org/0000-0002-3553-2491
Teemu J Murtola http://orcid.org/0000-0003-2932-9590

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
