## [Reviewer comments · BMJ Open]

ARTICLE DETAILS

TITLE (PROVISIONAL)	Randomized double-blind phase 3 clinical study testing impact of atorvastatin on prostate cancer progression after initiation of androgen deprivation therapy - study protocol
AUTHORS	Siltari, Aino; Riikonen, Jarno; Koskimäki, Juha; Pakarainen, Tomi; Ettala, Otto; Boström, Peter; Seikkula, Heikki; Kotsar, Andres; Tammela, Teuvo; Helminen, Mika; Raittinen, Paavo; Lehtimäki, Terho; Fode, Mikkel; Østergren, Peter; Borre, Michael; Rannikko, Antti; Marttila, Timo; Salonen, Arto; Ronkainen, Hanna; Löffeler, Sven; Murtola, Teemu J

VERSION 1 – REVIEW

REVIEWER	Singh, Prabhjot All India Institute of Medical Sciences
REVIEW RETURNED	10-Apr-2021

GENERAL COMMENTS	Comment: It's a well written study protocol. 1. Population included will be very heterogenous. There will be 2 subsets of patients 1. High risk non metastatic group 2. Metastatic group. Both behave differently. Time to develop castration resistance would be different . So, they should be analyzed separately. 2. “ All clinical decisions besides the study drug, e.g. use of early chemotherapy, abiraterone, or other drugs in adjunct with ADT will be up to the discretion of the attending clinician and allowed but will also be recorded in the trial database “ As per author, some patients will receive adjuvant treatment of Abiraterone/ chemo therapy/ enzalutamide along with ADT. So whole group would be heterogenous. Different treatment with ADT may be strong confounding factors. However, according to author they would be doing subgroup analysis. 3. “In every visit serum PSA, alkaline phosphatase, creatinine, and fasting glucose will be measured.” At the time of enrolment, only Alk Phos will be done. Atorvastatin usually affects liver enzymes i.e AST/ALT. which should be measured during follow up. 4. What modality will be used to diagnose metastasis at the time of recruitment? PSMA PET or bone scan alone?
--

REVIEWER	García-Cabezas, Sonia Reina Sofia University Hospital
REVIEW RETURNED	10-Apr-2021

GENERAL COMMENTS	This is a very interesting project, with the aim of improving the treatment of "de novo" metastatic prostate cancer or recurrent, a highly prevalent tumor with significant mortality. It seems hypercholesterolemia is a risk factor for the prognosis of prostate cancer, as has been reported by different studies. This is the first randomized placebo-controlled phase 3 clinical study on effects of atorvastatin on prostate cancer progression during ADT treatment. In my opinion it is a well justified and relevant project. It is a very complete study, which also includes a genetic and quality of life analysis. I just wanted to make a little comment. The dose of atorvastatin used is 80 mg, when generally the starting dose for patients with hypercholesterolaemia is lower. I understand that the possible benefit in prostate cancer has not been demonstrated with lower doses? In this sense, dose reduction in case of toxicity is not contemplated, but suspension.
---

REVIEWER	Dinneen, Eoin University College London Medical School, Department of Surgical and Interventional Sciences
REVIEW RETURNED	12-Apr-2021

GENERAL COMMENTS	Thank you to the authors for outlining their plan for this important and very interesting RCT proposal. This group clearly has great experience in this area of prostate cancer research as evidenced by their previous work, particularly reference 28. This study is the next step in this research story as they seek to influence patient outcomes in a high risk group and combine the RCT opportunity with basic science into statin and PCa interactions. I do not have a basic science background, so I am not well positioned to comment on their laboratory based plans. I will focus on the design of the clinical study, where I think the paper and possibly the design/size of the study would benefit from some further statistical considerations. On the whole, paper is extremely well-written, clear and compelling that this is an important research question to be addressed and that this avenue of investigation may well contribute some benefit to men in the future with this disease type and stage. On the whole, the methods are clear, inclusion + exclusion criteria seem appropriate. Have researchers considered stratifying randomisation for de novo metastatic vs. BCR following primary treatment, as these 2 represent very different patient populations and they should try to ensure equal distribution of these to both intervention and control arm. More importantly, I am confused by the interaction between 'Study Endpoint' and the Power calculation section. Page 7 line 58: 'Primary endpoint is the time to disease progression.' Then in Power calculations, the sample size offered does not refer to 'Time to Event' analysis, instead a treatment effects estimate difference between proportion of 17%. Furthermore, increasing sample size
--

	from 264 to 400 is a very large sample size increase and may not be necessary. What are the loss to follow-up assumptions that necessitate a 66% increase in sample size. I appreciate there is limited available RCTs to inform sample size calculation and certain assumptions are necessary, but this will have large implications for study design. If basing the power calculation off of Harhsmann et al. then analysis could be planned for approx 5-6 years after recruitment of the last (~300-400th) patient and in which case primary endpoint would be the proportion of patients in each arm having disease progression on ADT. Alternatively, 'time to event' analyses can use number of events and the rate at which they occur to be the basis for the power calculation. This data may not be readily available in the literature which again may necessitate some assumptions, but at the moment there is a discrepancy between the study primary endpoint and the rationale for the size of the study as it stands in the manuscript. Calculated properly, this may lead to an important increase or decrease in the size of the study which may decrease the cost and or increase likelihood of definitive finding. Looking at the study protocol, the same description of stats plan is given. Thank you. I look forward to seeing revision, which I will be certain to provide a positive review. I have indicated, for reasons above that the manuscript would benefit from stats review.
--	--

REVIEWER	Jones, Christopher Brighton and Sussex Medical School, Medical Statistics
REVIEW RETURNED	20-Sep-2021

GENERAL COMMENTS	The basis and rationale for the trial are well presented. I have some concerns about the statistical aspects, particularly sample size and proposed analyses. 1. Statistical methods section "Distribution balance of prostate cancer clinical characteristics (TNM stage at diagnosis, Gleason score, PSA at time of diagnosis) will be compared between the study arms using chi-square test for categorical variables and Mann-Whitney U-test or ANOVA for continuous variables depending on whether they follow normal distribution." These statistical tests of baseline variables are inappropriate in a randomised trial. It makes no sense to test null hypotheses of no difference between intervention groups at baseline, as due to the randomisation, we know these null hypotheses are true. Any observed differences would be due to chance. If these baseline variables are thought to be associated with the outcome, they should be included in modelling analyses for that outcome. "Recurrence-free time between the study arms will be compared using Kaplan-Meier curves and log-rank test. Risk of PSA relapse, as well as risk for occurrence of adverse tumor traits among men receiving atorvastatin compared to men receiving placebo will be estimated using Cox proportional hazards regression." Univariable analyses of outcomes (i.e. the log-rank test) should not be conducted as they can be misleading and are redundant when a multivariable model is fitted (i.e. in the Cox model). Adjusted Kaplan-Meier curves following Cox regression are more informative than unadjusted curves.
---

	"Statistical significance of the association between changes in serum lipid levels and occurrence of adverse tumor genomic traits will be estimated using ANOVA or Mann-Whitney U-test depending on the distribution normality." The non-parametric equivalent of an ANOVA (>2 groups) is Kruskal-Wallis. However, one way ANOVA or Kruskal-Wallis would likely be inappropriately simple - multivariable linear regression would likely be a more appropriate analysis. Analysis of QoL data is not mentioned in this section. It should be stated that a Statistical Analysis Plan will be developed and signed off prior to analysis. 2. The "Power calculation" section should be called "Sample size calculation". It's not immediately clear what numbers were entered in to the stated software to produce the sample size stated - please list all input parameters explicitly. State the expected drop out rate. This is likely to be high over such a long study. What is the minimally clinically important difference? It's not clear to me how updating the sample size calculation following an interim analysis will be useful. What will this analysis include? Descriptive summaries of each progression in each group may be appropriate, but might not provide much information to update the sample size calculation if not many participants' diseases have progressed. 3. Units of measurements for outcomes need to be stated where relevant. 4. "RedCap" (should be "REDCap") is mentioned but only when QoL data collection is discussed - presumably the trial database is a REDCap database - this should be clearer. 5. Some improvements to the English in the article could be made - non-exhaustive list of examples: "Importance of cholesterol is also supported by laboratory studies demonstrating importance of cholesterol for prostate cancer cell growth" -> "laboratory studies have demonstrated the importance of cholesterol for prostate cancer cell growth" "...statins appear to target also androgen..." -> "statins appear to also target androgen" "Besides cholesterol this pathway produces also isoprenoid proteins" -> "Besides cholesterol, this pathway also produces isoprenoid proteins" "The drug rations and capsules containing atorvastatin and placebo will be of identical outlook" / "Placebo capsules will be identical in outlook as atorvastatin capsules" -> "Atorvastatin and placebo capsules will be identical in appearance." ("outlook" -> "appearance" throughout).
--	---

	"Number of the remaining capsules" -> "Number of capsules remaining" "alfa" -> "alpha" 6. The study is described as "double-blinded" but this is not specific. It's better to describe it as blinded and then specifically state who will be blind (this is mostly stated). Will the analysis be conducted blind? 7. No mention is made of showing participant flow through the study in a CONSORT diagram. 8. What software was used to produce the randomisation list? 9. What software will be used for analysis?
--	--

VERSION 1 – AUTHOR RESPONSE

Reviewer: 1

Dr. Prabhjot Singh, All India Institute of Medical Sciences

Comments to the Author:

Comment:

It's a well written study protocol.

1. Population included will be very heterogenous. There will be 2 subsets of patients

1. High risk non metastatic group 2. Metastatic group.

Both behave differently. Time to develop castration resistance would be different . So, they should be analyzed separately.

Response: Thank you for your comments. We are aware that our subsets of patients might behave differently e.g. time to formation of our primary endpoint might be different. Our protocol includes a planned subgroup analysis stratified by metastatic status at time of starting ADT. Of note, our target group is similar to STAMPEDE trial which has turned out to be very successful.

2. " All clinical decisions besides the study drug, e.g. use of early chemotherapy, abiraterone, or other drugs in adjunct with ADT will be up to the discretion of the attending clinician and allowed but will also be recorded in the trial database "

As per author, some patients will receive adjuvant treatment of Abiraterone/ chemotherapy/ enzalutamide along with ADT. So whole group would be heterogenous. Different treatment with ADT may be strong confounding factors. However, according to author they would be doing subgroup analysis.

Response: We appreciate reviewers comment and concern about the heterogenous of our subjects and as mentioned already above, we have planned to perform sub-analysis and sensitivity analysis in patients with similar clinical characteristics. As we plan to recruit up to 400 patients, we trust randomization to make treatment characteristics similar between the study arms.

3. "In every visit serum PSA, alkaline phosphatase, creatinine, and fasting glucose will be measured." At the time of enrolment, only Alk Phos will be done. Atorvastatin usually affects liver enzymes i.e AST/ALT. which should be measured during follow up.

Response: Indeed, in each visit PSA, alkaline phosphatase, creatinine, serum creatine kinase, alanine aminotransferase, fasting blood glucose, and lipid levels are measured. We have now corrected this in our manuscript and in Table 2.

4. What modality will be used to diagnose metastasis at the time of recruitment? PSMA PET or bone scan alone?

Response: As our trial is pragmatic, and includes several centers with varying imaging possibilities, the choice of imaging modality at diagnosis is up to the clinician.

Reviewer: 2

Dr. Sonia García-Cabezas, Reina Sofia University Hospital

Comments to the Author:

This is a very interesting project, with the aim of improving the treatment of "de novo" metastatic prostate cancer or recurrent, a highly prevalent tumor with significant mortality.

It seems hypercholesterolemia is a risk factor for the prognosis of prostate cancer, as has been reported by different studies.

This is the first randomized placebo-controlled phase 3 clinical study on effects of atorvastatin on prostate cancer progression during ADT treatment.

In my opinion it is a well justified and relevant project. It is a very complete study, which also includes a genetic and quality of life analysis.

I just wanted to make a little comment.

The dose of atorvastatin used is 80 mg, when generally the starting dose for patients with hypercholesterolaemia is lower. I understand that the possible benefit in prostate cancer has not been demonstrated with lower doses? In this sense, dose reduction in case of toxicity is not contemplated, but suspension.

Response: We are grateful for reviewer's comments. We agree that the starting dose of atorvastatin for its primary indication of treatment of hypercholesterolemia or primary prevention of cardiovascular disease is high. However, when atorvastatin is used to affect cancer, we expect the dose needs to be higher than for the primary indication. Further, we have shown in our previous clinical drug trial that prostate cancer patients tolerated atorvastatin well at 80 mg dose (Murtola et al. Eur Urol. 2018) and the dose was sufficient for atorvastatin to be detected in prostatic tissue after prostatectomy (Knuutila et al. Prostate 2019).

The study drug is a single capsule containing either the full dose 80 mg atorvastatin or placebo. Therefore, dose reductions are not possible.

Reviewer: 3

Mr. Eoin Dinneen, University College London Medical School

Comments to the Author:

Thank you to the authors for outlining their plan for this important and very interesting RCT proposal.

This group clearly has great experience in this area of prostate cancer research as evidenced by their previous work, particularly reference 28. This study is the next step in this research story as they seek to influence patient outcomes in a high risk group and combine the RCT opportunity with basic science into statin and PCa interactions.

I do not have a basic science background, so I am not well positioned to comment on their laboratory based plans.

I will focus on the design of the clinical study, where I think the paper and possibly the design/size of the study would benefit from some further statistical considerations.

On the whole, paper is extremely well-written, clear and compelling that this is an important research question to be addressed and that this avenue of investigation may well contribute some benefit to men in the future with this disease type and stage.

Response: Thank you for your kind comments.

On the whole, the methods are clear, inclusion + exclusion criteria seem appropriate. Have researchers considered stratifying randomisation for de novo metastatic vs. BCR following primary treatment, as these 2 represent very different patient populations and they should try to ensure equal distribution of these to both intervention and control arm.

Response: In this study, randomization is done at the time of study drug production. Each study drug package gets a specific code. When a patient is recruited to the study, he gets the next available study code and the blinded treatment assigned to that particular code. Therefore, randomization cannot be influenced at the time of recruitment, hence stratification by clinical characteristics is not possible.

We are aware that heterogeneity between subjects is expected. Nevertheless, in study population of this size we expect randomization to ensure equal distribution of clinical characteristics between treatment arms. Additionally, we do include planned subgroup analyses stratified by relevant clinical characteristics.

More importantly, I am confuse by the interaction between 'Study Endpoint' and the Power calculation section. Page 7 line 58: 'Primary endpoint is the time to disease progression.' Then in Power calculations, the sample size offered does not refer to 'Time to Event' analysis, instead a treatment effects estimate difference between proportion of 17%. Furthermore, increasing sample size from 264 to 400 is a very large sample size increase and may not be necessary. What are the loss to follow-up assumptions that necessitate a 66% increase in sample size. I appreciate there is limited available RCTs to inform sample size calculation and certain assumptions are necessary, but this will have large implications for study design. If basing the power calculation off of Harshmann et al. then analysis could be planned for approx 5-6 years after recruitment of the last (~300-400th) patient and in which case primary endpoint would be the proportion of patients in each arm having disease progression on ADT. Alternatively, 'time to event' analyses can use number of events and the rate at which they occur tbe the basis for the power calculation. This data may not be readily available in the literature which again may necessitate some assumptions, but at the moment there is a discrepancy between the study primary endpoint and the rationale for the size of the study as it stands in the manuscript. Calculated properly, this may lead to an important increase or decrease in the size of the study which may decrease the cost and or increase likelihood of definitive finding. Looking at the study protocol, the same description of stats plan is given.

Response: We agree on the above points. Our power calculation is indeed limited by being based on retrospective data as RCT data is unavailable. This is the reason why we include an interim analysis after first 100 patients have been recruited and followed for a minimum of one year. This interim analysis will give us an idea of the effect size to be expected in randomized setting. This information is used to validate or revise our sample size.

Furthermore, our statistician did some tentative power calculation based on Harsmann et al. where difference between statin users and non-users to development of castration resistance were 58% vs. 75% at the timepoint of 5 years. Thus, if HR lowers to around 0.65 in statin group compared to placebo, out power is sufficient (see figure 2). However, as mentioned we will recalculate our sample size when interim analysis is done.

Thank you.

I look forward to seeing revision, which I will be certain to provide a positive review. I have indicated, for reasons above that the manuscript would benefit from stats review.

Reviewer: 4

Dr. Christopher Jones, Brighton and Sussex Medical School

Comments to the Author:

The basis and rationale for the trial are well presented. I have some concerns about the statistical aspects, particularly sample size and proposed analyses.

1. Statistical methods section

"Distribution balance of prostate cancer clinical characteristics (TNM stage at diagnosis, Gleason score, PSA at time of diagnosis) will be compared between the study arms using chi-square test for categorical variables and Mann-Whitney U-test or ANOVA for continuous variables depending on whether they follow normal distribution."

These statistical tests of baseline variables are inappropriate in a randomised trial. It makes no sense to test null hypotheses of no difference between intervention groups at baseline, as due to the randomisation, we know these null hypotheses are true. Any observed differences would be due to chance. If these baseline variables are thought to be associated with the outcome, they should be included in modelling analyses for that outcome.

Response: Indeed this is true. The baseline characteristics are not expected to be associated with the outcome in a randomized trial. Therefore, we have removed mention of statistical testing in this context. Instead we describe how distribution of baseline characteristics will be shown in a table.

"Recurrence-free time between the study arms will be compared using Kaplan-Meier curves and log-rank test. Risk of PSA relapse, as well as risk for occurrence of adverse tumor traits among men receiving atorvastatin compared to men receiving placebo will be estimated using Cox proportional hazards regression."

Univariable analyses of outcomes (i.e. the log-rank test) should not be conducted as they can be misleading and are redundant when a multivariable model is fitted (i.e. in the Cox model). Adjusted Kaplan-Meier curves following Cox regression are more informative than unadjusted curves.

Response: In RCT setting, the Kaplan-Meier (KM) estimates are not, by design, prone to bias due to treatment-selection. That is, weighting / adjusting Kaplan-Meier estimates in the context of RCT should not be necessary, assuming successful randomization. Should any baseline imbalances occur, we will then use adjusted Cox proportional hazards model.

"Statistical significance of the association between changes in serum lipid levels and occurrence of adverse tumor genomic traits will be estimated using ANOVA or Mann-Whitney U-test depending on the distribution normality."

The non-parametric equivalent of an ANOVA (>2 groups) is Kruskal-Wallis. However, one way ANOVA or Kruskal-Wallis would likely be inappropriately simple - multivariable linear regression would likely be a more appropriate analysis.

Response: We have expanded this section to include more details about the statistical analysis.

Analysis of QoL data is not mentioned in this section.

Response: We have added QoL analysis details into the manuscript.

It should be stated that a Statistical Analysis Plan will be developed and signed off prior to analysis.

Response: We have added that a Statistical Analysis Plan will be written, signed off and published before the analysis.

2. The "Power calculation" section should be called "Sample size calculation".

Response: We have now changed the name of the section.

It's not immediately clear what numbers were entered in to the stated software to produce the sample size stated - please list all input parameters explicitly.

Response: Calculation was made with assumption that PSA relapse after 5 y is 75% in placebo and 58% in statin group as Harsmann et al. showed. In calculations power was set to 80% and alpha to 0.05.

State the expected drop-out rate. This is likely to be high over such a long study. What is the minimally clinically important difference?

Response: We assume 10 % drop-out rate in each study arms. Clinical importance of variables like time to castration resistance is subjective, our estimate is that 3 months longer time to castration resistance would be clinically significant. This is now mentioned in the text.

It's not clear to me how updating the sample size calculation following an interim analysis will be useful. What will this analysis include? Descriptive summaries of each progression in each group may be appropriate, but might not provide much information to update the sample size calculation if not many participants' diseases have progressed.

Response: As this is the first randomized clinical trial investigating the topic, sample size calculation is based on retrospective studies prone to bias. Therefore, it is important to check our sample size based on randomized data. We chose to do this after 100 patients have been recruited and all have been followed for minimum of 1 year. This means that most men have been followed for longer than that as recruitment of 100 men takes time. Our target group is men with most aggressive type of prostate cancer, many with metastatic disease. Therefore, number of men meeting the primary endpoint of castration resistance is expected to be high already at this time point, and we see the interim analysis as important tool to check the adequacy of the sample size calculation.

3. Units of measurements for outcomes need to be stated where relevant.

Response: We have now added units wherever relevant.

4. "RedCap" (should be "REDCap") is mentioned but only when QoL data collection is discussed - presumably the trial database is a REDCap database - this should be clearer.

Response: We have clarified our eCFR database wherever needed and corrected the spelling of REDCap.

5. Some improvements to the English in the article could be made - non-exhaustive list of examples: "Importance of cholesterol is also supported by laboratory studies demonstrating importance of cholesterol for prostate cancer cell growth" -> "laboratory studies have demonstrated the importance of cholesterol for prostate cancer cell growth"

"...statins appear to target also androgen..." -> "statins appear to also target androgen"

"Besides cholesterol this pathway produces also isoprenoid proteins" -> "Besides cholesterol, this pathway also produces isoprenoid proteins"

"The drug rations and capsules containing atorvastatin and placebo will be of identical outlook" / "Placebo capsules will be identical in outlook as atorvastatin capsules" -> "Atorvastatin and placebo capsules will be identical in appearance."
("outlook" -> "appearance" throughout).

"Number of the remaining capsules" -> "Number of capsules remaining"

"alfa" -> "alpha"

Response: We are grateful for reviewer's corrections and have now added improvements to the language of the manuscript.

6. The study is described as "double-blinded" but this is not specific. It's better to describe it as blinded and then specifically state who will be blind (this is mostly stated). Will the analysis be conducted blind?

Response: We have now added statement to abstract about blinding and it is more precisely explained in manuscript page 7, rows 14-16. Blinding is lifted after recruitment target is achieved and all patients have been followed-up for at least one year, thus, all other analysis will be performed unblinded except interim analysis after 100 patients have reach out primary endpoint.

7. No mention is made of showing participant flow through the study in a CONSORT diagram.

Response: We have created Figure 1 which includes study flow with inclusion and exclusion criteria. We will form more specific study flow diagram with n-values for each step when trials primary endpoint is reported. Figure 1 is mentioned in the beginning of the Study setting section in page 5.

8. What software was used to produce the randomisation list?

Response: Randomization list was created by randomly sequencing treatment arm (1 or 0, atorvastatin or placebo) by assigning random numbers between 1 and 400 for each treatment in Excel.

9. What software will be used for analysis?

Response: We have now added statement that SPSS, Stata and R softwares will be used for statistical analysis..

VERSION 2 – REVIEW

REVIEWER	Singh, Prabhjot All India Institute of Medical Sciences
REVIEW RETURNED	31-Dec-2021

GENERAL COMMENTS	All queries have been answered
--------------------------------

REVIEWER	Jones, Christopher Brighton and Sussex Medical School, Medical Statistics
REVIEW RETURNED	17-Dec-2021

GENERAL COMMENTS	My previous comments have been resolved.
--